# The delayed kinetics of Myddosome formation explains why amyloid-beta aggregates trigger Toll-like receptor 4 less efficiently than lipopolysaccharide

**Bing Li[1,2†], Prasanna Suresh[1,2†], Jack Brelstaff[3], Shekhar Kedia[1,2], Clare E Bryant[3]\*, David Klenerman[1,2]\***

[1]Department of Chemistry, University of Cambridge, Cambridge, United Kingdom; [2]Cambridge Dementia Research Centre, Clifford Allbutt Building, Cambridge Biomedical Campus, Cambridge, United Kingdom; [3]Department of Medicine, Addenbrooke's Hospital, Cambridge Biomedical Campus, Cambridge, United Kingdom

**\*For correspondence:**
ceb27@cam.ac.uk (CEB);
dk10012@cam.ac.uk (DK)

[†]These authors contributed equally to this work

**Competing interest:** The authors declare that no competing interests exist.

**Abstract** The Myddosome is a key innate immune signalling platform. It forms at the cell surface and contains MyD88 and IRAK proteins which ultimately coordinate the production of pro-inflammatory cytokines. Toll-like receptor 4 (TLR4) signals via the Myddosome when triggered by lipopolysaccharide (LPS) or amyloid-beta (Aβ) aggregates but the magnitude and time duration of the response are very different for reasons that are unclear. Here, we followed the formation of Myddosomes in live macrophages using local delivery of TLR4 agonist to the cell surface and visualisation with 3D rapid light sheet imaging. This was complemented by super-resolution imaging of Myddosomes in fixed macrophages to determine the size of the signalling complex at different times after triggering. Myddosomes formed more rapidly after LPS than in response to sonicated Aβ 1–42 fibrils (80 vs 372 s). The mean lifetimes of the Myddosomes were also shorter when triggered by LPS compared to sonicated Aβ fibrils (170 and 220 s), respectively. In both cases, a range of Myddosome of different sizes (50–500 nm) were formed. In particular, small round Myddosomes around 100 nm in size formed at early time points, then reduced in proportion over time. Collectively, our data suggest that compared to LPS the multivalency of Aβ fibrils leads to the formation of larger Myddosomes which form more slowly and, due to their size, take longer to disassemble. This explains why sonicated Aβ fibrils results in less efficient triggering of TLR4 signalling and may be a general property of protein aggregates.

## eLife assessment

This **important** study uses a novel light sheet imaging technique to investigate how different TLR4 agonists regulate Myddosome formation. The data showing that LPS and A-beta can control the kinetics and size of Myddosome assembly are **compelling**. This paper should be of substantial interest to the innate immunity field.

## Introduction

Signalling through Toll-like receptor 4 (TLR4) in response to bacterial lipopolysaccharide (LPS) potently drives the production of pro-inflammatory cytokines, such as tumor necrosis factor α (TNFα) and the type I interferon (IFN) IFNβ, during Gram-negative bacterial infection. It is now clear that

TLR4 activation is important in many diseases with, for example, its triggering by small aggregates of proteins, such as amyloid-beta (Aβ), thought to be important in the initiation and development of Alzheimer's disease (AD) (**Okamura et al., 2001**; **Reed-Geaghan et al., 2009**). Two rare genetic variants in the APOE or TREM2 genes are associated with AD and these proteins both can modulate TLR4 signalling. Importantly, TLR4 priming is required for full activation of the NLRP3 inflammasome which is also thought to play an important role in the aetiology of many of AD (**Juliana et al., 2012**).

Signalling through TLR4 utilises four adaptor proteins: Myelin and lymphocyte protein (Mal)/ (TIR) domain-containing adaptor protein (Tirap), Myeloid differentiation primary response 88 (MyD88), Translocating chain-associated membrane protein (Tram), and Toll/interleukin-1 receptor domain-containing adaptor protein inducing interferon beta (Trif) which form two macromolecular signalling complexes: the Myddosome and the Triffosome, respectively (**Gay et al., 2014**). The Myddosome consists of MyD88 and IRAK-2 and -4 kinases. In a crystal structure of the Myddosome there are six MyD88 molecules, four IRAK4, and four IRAK2 subunits (**Lin et al., 2010**). Our biophysical analysis within cells, also identifies complexes with six MyD88, but other stoichiometries are present (**Latty et al., 2018**; **Moncrieffe et al., 2020**). Our previous work shows that low doses of small Aβ-42 oligomers (at concentrations of these proteins found in the cerebrospinal fluid of AD patients) primes TLR4 signalling to induce sustained cytokine production, through Myddosome formation (**Hughes et al., 2020**), which is a pattern of signalling that is very different to that seen when TLR4 is triggered by LPS. It is unclear how such different patterns of TLR4 signalling are induced by protein aggregates in comparison to LPS. The Aβ aggregates are much larger than LPS and potentially multivalent so it is possible they could bind multiple TLR4 simultaneously potentially sterically hindering the successful formation of Myddosomes and hence altering the signalling efficiency.

Studying live Myddosome formation in cells is technically challenging for many reasons including labelling the key proteins, the resolution of imaging required to identify individual proteins and the fact that TLR4 traffics rapidly away from the cell surface. Our previous work used fluorescently tagged MyD88 in total internal reflection fluorescence (TIRF) microscopy to follow Myddosomes formation in macrophages (**Latty et al., 2018**). We found that Myddosomes assemble within minutes of TLR4 stimulation and that they contained six MyD88 molecules although larger Myddosomes could form. Lipid partial agonists at TLR4 stimulate slower formation of a smaller number of Myddosomes. Only a small number of Myddosomes need to be formed on the cell surface for full cellular signalling to occur, and we proposed that the difference between full and partial agonism is determined by Myddosome number, size, and the speed of their formation. Elegant work using TIRF microscopy supported our hypothesis when studying Myddosome formation in response to triggering of the interleukin 1 receptor in mouse lymphoma cells (**Deliz-Aguirre et al., 2021**). Here, small reversible Myddosomes formed but signalling required the formation of stable larger Myddosomes with four or more MyD88. Both these previous imaging studies on Myddosomes used TIRF microscopy. This technique limits the volume that can be visualised to 200 nm above the coverslip such that only Myddosomes formed on the bottom cell membrane can be seen so any Myddosomes formed away from this field of view will be lost. It is also hard to calculate the lifetime of assembled Myddosomes with TIRF because the loss of the Myddosome fluorescence signal could be attributed to either disassembly or trafficking of the complex away from the bottom cell membrane. Confocal microscopy images cells in 3D, but its low imaging speed and out of plane photobleaching means this technique is not capable of repetitive live cell imaging of low-level fluorescence signals. Signalling through TLR4 is fast so to visualise early events, such as Myddosome formation, ideally signalling should be initiated at a defined time, such as by controllably delivering agonists to the surface of a single cell and then immediately imaging the cellular response.

Here, we combined light sheet microscopy with local agonist delivery, via a nanopipette, to determine why Aβ aggregates trigger such a distinctive pattern of TLR4 signalling compared to LPS. Light sheet 3D scanning, by scanning the objective or sample stage, can scan the whole-cell sample with relative high speed (single cell scanning only takes 2 s). The use of a nanopipette, will accurately deliver individual molecules to a defined position on the cell surface. We called this method local-delivery selective-plane illumination microscopy (ldSPIM) (**Li et al., 2021**). ldSPIM uses a nanopipette to control the position from which the molecules are delivered and the lightsheet microscopy to follow Myddosome formation in live macrophages following TLR4 triggering by LPS or sonicated Aβ fibrils.

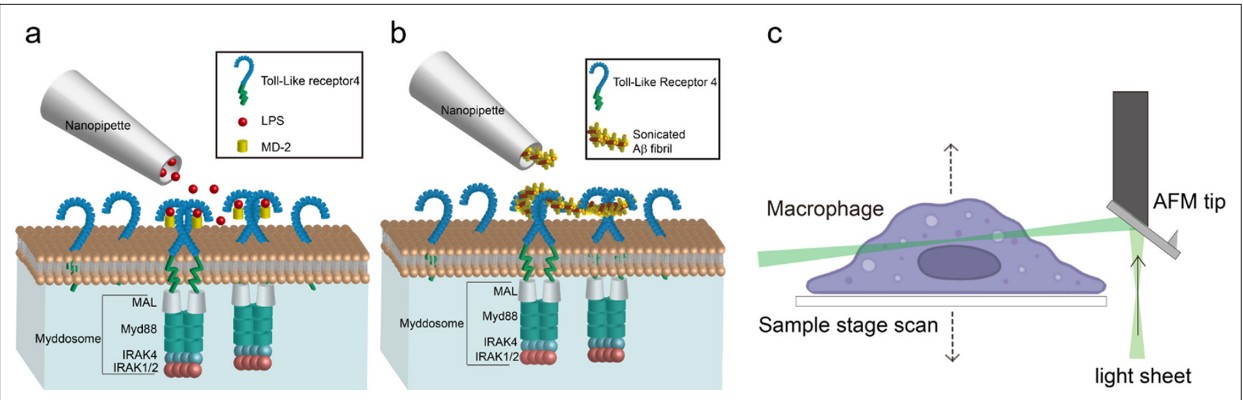

**Figure 1.** Schematic of local-delivery selective plane illuminatiion microscopy. (**a, b**) Schematic of nanopipette delivered triggering of Toll-like receptor 4 (TLR4) and Myddosome formation with lipopolysaccharide (LPS) or amyloid-beta (Aβ) aggregates. The LPS is delivered by nanopipette to the macrophage surface. When LPS binds to TLR4, it will trigger the dimerisation of TLR4, which activates the recruitment of Mal, MyD88, and IRAK4&2 to form the Myddosome. Unstimulated TLR4 remains monomeric on the cell surface. (**c**) Schematic of Atomic Force Microscope (AFM) cantilever light sheet 3D scan. The light sheet is reflected by an AFM cantilever to the target cell. Control by a piezo, the sample stage is moving up and down to achieve cell scanning.

The online version of this article includes the following figure supplement(s) for figure 1:

**Figure supplement 1.** Transmission Electron Microscopy images of amyloid-β 42 following sonication of incubated fibrils.

**Figure supplement 2.** Nanopipette delivery of sonicated amyloid beta fibrils.

## Results

### IdSPIM captures the precise formation kinetics of all Myddosomes within a cell

We used IdSPIM to follow TLR4 activation and quantitatively characterise the formation of the Myddosome as shown schematically in *Figure 1*. A pipette with a diameter of 200 nm for LPS or 800 nm for sonicated Aβ1–42 fibrils was positioned 3 μm above the cell surface, and a pressure pulse for 5 s was used for delivery of the TLR4 agonist onto the surface of MyD88$^{-/-}$ immortalised bone marrow-derived macrophages (iBMDMs) transduced with MyD88-YFP. The assembly of MyD88 oligomers after TLR4 activation was then visualised in live cells using 3D light sheet scanning.

Compared with LPS, the sonicated Aβ fibrils consist of larger aggregates, so it is important to ensure that the nanopipette is capable of delivering the fibrils smoothly without blockage. We firstly characterised the Aβ fibrils using electron microscopy, showing that the majority was less than 100 nm in length (*Figure 1—figure supplement 1*). To test the macromolecule delivery ability of the nanopipette, the sonicated Aβ fibrils (4 μM total monomer concentration) were first tagged with a 1:1000 dilution of Amytracker 680, a small molecule dye which specifically binds to aggregates; this was then delivered to the macrophage surface. *Figure 1—figure supplement 2a, b* shows the deposition of Amytracker tagged aggregates at the cell membrane demonstrating reliable delivery of sonicated fibrils to the cell membrane.

Macrophages were activated by both LPS and sonicated Aβ fibrils to form Myddosomes (*Figure 2*). A typical result showed that 450 s after triggering with sonicated Aβ fibrils, the first Myddosome was formed, with continuous assembly of new complexes for the next 1000s. The fibrils are detected on the cell surface immediately after dosing, so time taken to visualise a Myddosome accurately records the kinetics of Myddosome formation. In the reconstructed 3D scan image of the whole cell, any Myddosomes formed in the total cell volume can be seen (*Figure 3*). The spatial distribution and movement of each individual Myddosome were tracked and shown in 3D volume rendered images, which was projected using the brightest point method (*Figure 3*). Compared with TIRF, 3D light sheet imaging is able to follow all Myddosomes formed, allowing for the calculation of the lifetime of each individual Myddosome, and the total number of Myddosomes assembled within a certain time period post triggering precisely. Our previous studies have demonstrated that when TLR4 is triggered by various agonists, the response speed and magnitude of response vary; therefore, we measured the structure and dynamics of Myddosome assembly following triggering by LPS and sonicated Aβ fibrils,

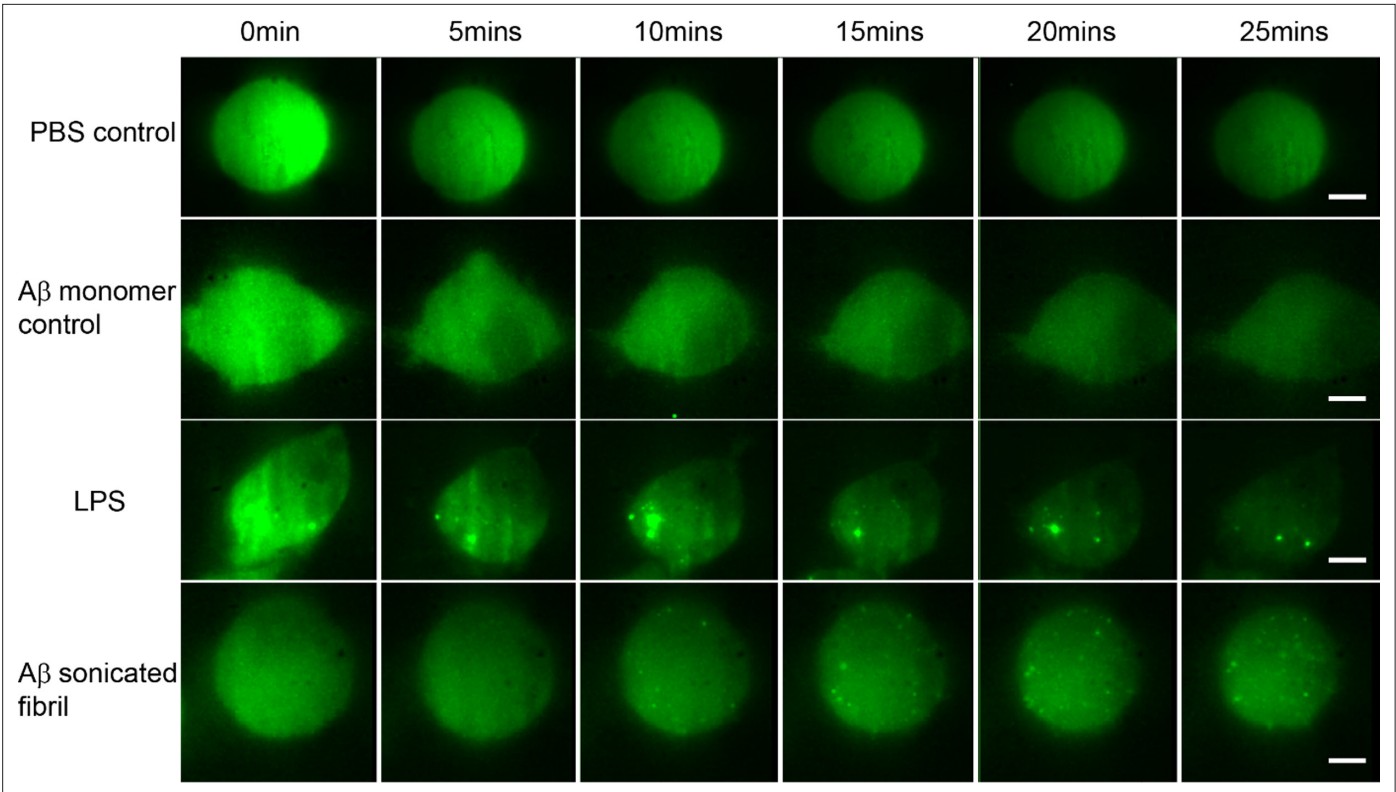

**Figure 2.** Montage showing the time series of Myddosome assembly (small bright puncta) for different stimulations. (**a**) PBS (phosphate-buffered saline) control: no Myddosomes formed when PBS buffer is delivered. (**b**) Amyloid-beta (Aβ) monomer control also showing no Myddosomes formed. (**c**) Myddosome formation triggered by lipopolysaccharide (LPS). (**d**) Myddosome formation triggered by sonicated Aβ fibrils. The scale bar is 5 µm.

The online version of this article includes the following figure supplement(s) for figure 2:

**Figure supplement 1.** Overview of 2D tracking analysis.

and analysed the data to determine the following parameters: (1) the first time point when Myddosomes formed, (2) the Myddosome lifetime, (3) the total number of oligomers formed within 30 min of triggering, (4) the size and shape distribution of assembled Myddosomes and (5) the variation of MyD88 oligomer size with stimulation time.

## Aβ triggers delayed Myddosome compared to LPS

First, the time point when the first Myddosome formed was measured. PBS and Aβ monomers were first delivered to macrophages as controls, which resulted in no triggering (*Figure 2*). Then LPS and sonicated Aβ fibrils were delivered to macrophages for 5 s and imaged by 3D light sheet scan. LPS-triggered Myddosomes started forming within 5 min after local delivery. Myddosomes formed after triggering by sonicated Aβ fibrils first appeared after a significantly longer time, ranging from 200 to 1000s (*Figure 4*). On average the LPS-triggered Myddosome first formed at 80 s after delivery and the Aβ-triggered Myddosome first formed at 370 s after delivery, with LPS stimulation leading to a significantly faster time to first formation than sonicated Aβ fibrils (unpaired two-sided Student's *t*-test, p = 3.33E−8).

## The lifetime of Aβ-triggered Myddosomes is extended compared to LPS

Myddosome lifetimes were determined after obtaining trajectories for each puncta using a particle tracking algorithm. The lifetime of both LPS and sonicated Aβ fibrils triggered Myddosomes vary significantly and have a large population below 250 s (*Figure 5*). The cumulative lifetimes plot (*Figure 5c*) shows that Aβ-triggered Myddosomes had a higher proportion of longer lived Myddosomes than LPS.

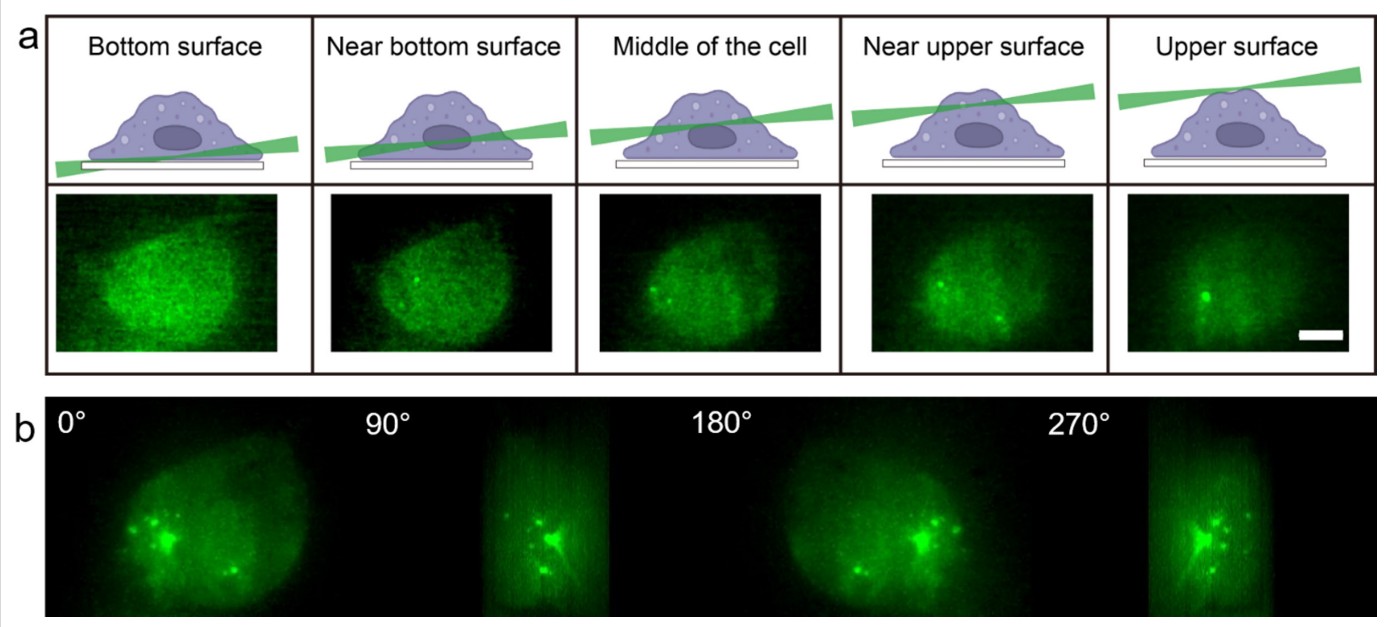

**Figure 3.** Myddosome visualisation throughout full cell volume enabled by light sheet microsope. (**a**) 3D light sheet scanning of a live macrophage forming Myddosomes upon activation. The sheet scanning starts from the bottom surface of the cell and ends at the top surface. Z-stacks were acquired for each cell, consisting of 100 z-slices with 200 nm spacing. (**b**) Images were rendered by 3D projection (brightest point method). Myddosomes could be visualised in the whole-cell volume by rotating the 3D rendered imaged around the y-axis.

To calculate the lifetime accurately, it was important to determine that the YFP fluorescence signal of MyD88 puncta did not bleach during 3D scanning. Macrophages with LPS-triggered Myddosomes were therefore fixed and imaged using the same laser power and same scanning time as used for 3D imaging. The time taken for the fixed puncta to bleach was much longer compared to the timescales that the Myddosomes were degrading in live cells after stimulation (*Figure 5—figure supplement 1*).

The lifetime distributions also provide information about degradation kinetics (*Tinoco and Gonzalez, 2011*). The distributions were normalised so that the integral over all positive values equalled 1, and the probability distribution function ($P\left(\tau\right)$) fit assuming a two-step degradation (*Figure 6*):

$$P\left(\tau\right) = \frac{k_1 k_2}{k_2 - k_1}\left(e^{-k_1\tau} - e^{-k_2\tau}\right)$$

The mean lifetime $< \tau >$ can be determined by:

$$< \tau > = \int_0^\infty \tau P\left(\tau\right) d\tau = \frac{k_1 + k_2}{k_1 k_2}$$

The mean Myddosome lifetimes when triggered by LPS or sonicated Aβ fibrils were 162 and 182 s, respectively.

## Myddosomes predominantly remain at the cell membrane

We examined the number of Myddosomes that stay on the cell surface or are internalised to the cytosol, assuming that signalling competent Myddosomes only form at the membrane after triggering with LPS or Aβ fibrils. By filtering to only include Myddosome trajectories which form at the membrane, any localisations in the cytoplasm are due to Myddosomes which have internalised after assembly (*Figure 7*). In both LPS- and Aβ-triggered cells, following an initial increase after triggering, the population of membrane localised Myddosomes remained stable, and higher than the number of cytoplasmic Myddosomes, suggesting there most Myddosomes did not undergo internalisation in the first 30 min after stimulation.

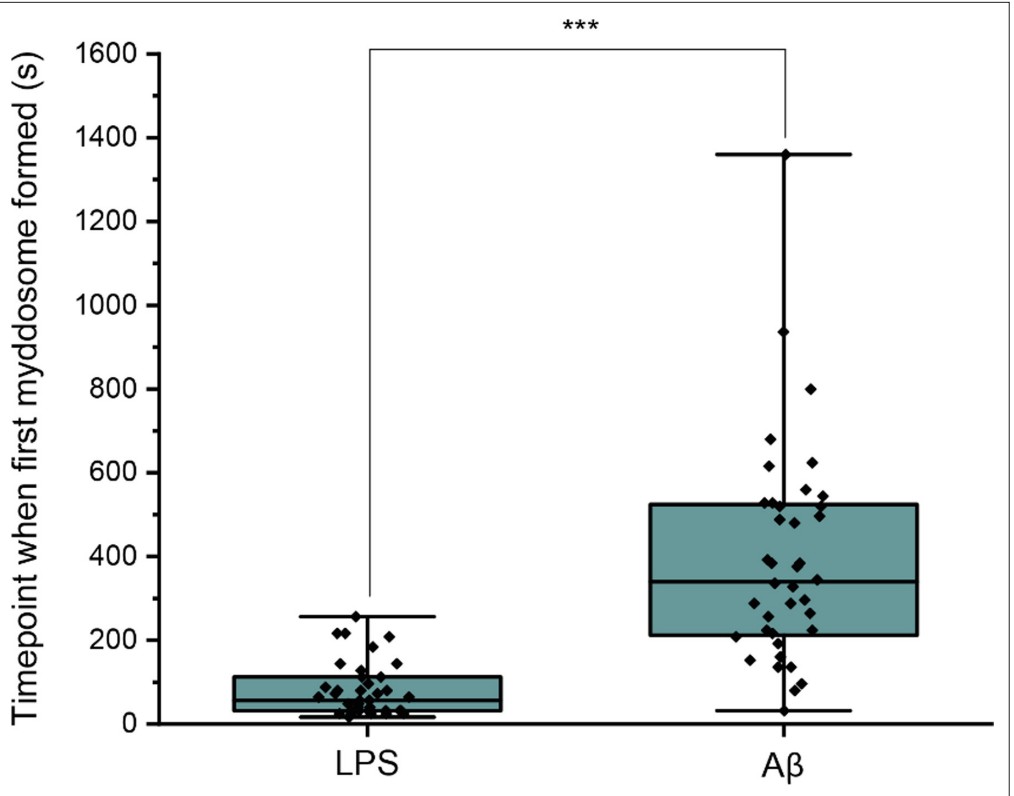

**Figure 4.** Differences in Myddosome initial formation following LPS/Aβ stimulation. (**a**) MyD88-YFP transduced iBMDMs were stimulated with 1 μg/ml lipopolysaccharide (LPS) or 4 μM total monomer of sonicated amyloid-beta (Aβ) fibrils delivered from a nanopipette for 30 min. The time point when the first Myddosome formed was marked (*n* = 77) across five biological replicates for each stimulation. The p-values are based on unpaired two-sided Student's *t*-test.

The online version of this article includes the following source data for figure 4:

**Source data 1.** Timepoints following stimulation at which Myddosome formation starts.

## Larger Myddosomes are formed in response to Aβ than LPS

We used direct stochastic optical reconstruction microscopy (dSTORM) to visualise and quantify the size of LPS and sonicated Aβ aggregates triggered Myddosomes after fixation. In this case, the agonists were applied in the bath. As the MyD88-YFP could not be super-resolved directly, an Alexa Fluor 647-conjugated anti-GFP antibody was used to label MyD88-YFP for super-resolution imaging (*Figure 8*). The merged channel overlaps the 488 and 641 nm images, showing the colocalisation of MyD88-YFP and the super-resolved Myddosomes. The control cells (MyD88-YFP without stimulation) showed no formation of Myddosomes in the 488 nm channel, and no overlapping signal in the merged channel. The LPS and Aβ-triggered cells had punctate signal in the 488 nm channel, with some of these puncta displaying colocalisation with the super-resolved signal in the 641 nm channel. We used co-localisation to identify the real Myddosomes from non-specific signal. The size (calculated from the 1D Full Width Half Maxima) of Myddosome puncta formed at different time points after LPS and sonicated Aβ fibrils stimulation are shown in *Figure 9*. The Myddosomes formed by sonicated Aβ fibrils stimulation are larger than those formed by LPS stimulation at all times, especially at early times.

We then analysed our super-resolution data in more detail, looking specifically at the shape factors of each Myddosome (*Figure 10*). The shape factor, which is calculated using the perimeter and area of a cluster using the ratio $\frac{4\pi A}{p^2}$ , can take a value between 0 and 1 depending on the circularity of the cluster. We started by measuring the shape factor distributions for LPS and Aβ at an early time point (30 min following stimulation). For both conditions, a bimodal distribution was observed (*Figure 10a*), consisting of a population of highly circular MyD88 clusters, as well as a more variable population of less circular clusters. By comparing this distribution to the area of the clusters, we observed that this

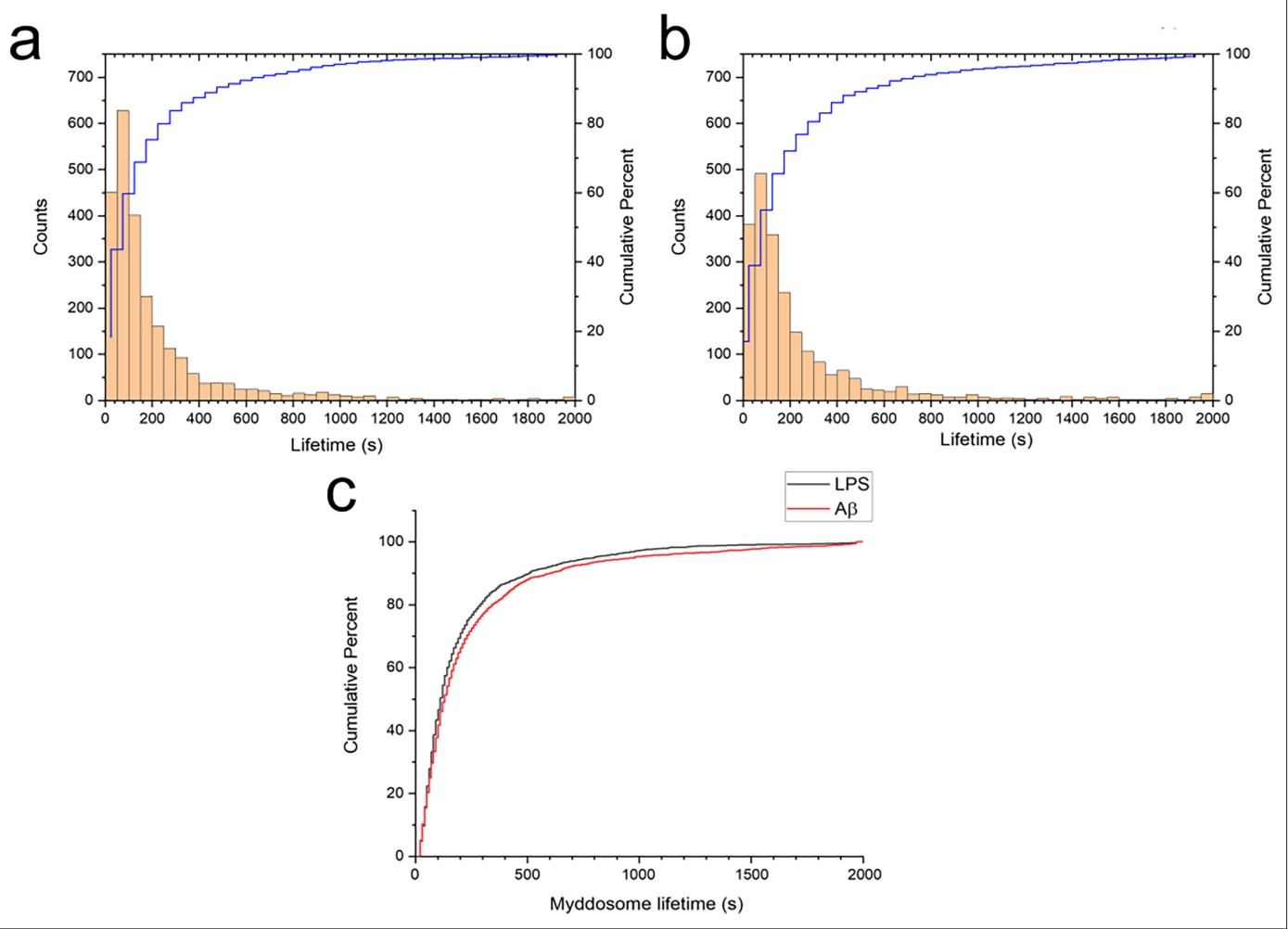

**Figure 5.** Myddosome puncta lifetimes following stimulation. Histogram and cumulative distribution of MyD88 puncta lifetimes following nanopipette delivery of (**a**) lipopolysaccharide (LPS) (1 µg/ml) or (**b**) sonicated amyloid-beta (Aβ) fibrils (4 µM total monomer). (**c**) Cumulative lifetime distributions after LPS and Aβ stimulation overlaid. The difference between the two lifetime distributions was significant (Kolmogorov–Smirnov test, p < 0.0001).

The online version of this article includes the following source data and figure supplement(s) for figure 5:

**Source data 1.** Myddosome puncta lifetimes following stimulation.

**Figure supplement 1.** Cells were fixed following lipopolysaccharide (LPS) triggering, and then imaging using the same parameters and microscope as previously described (Materials and methods: Live cell scanning and 3D reconstruction).

---

population clusters with a shape factor close to 1 were mostly very small, with an area below 0.01 µm² (equivalent to a diameter below approximately 100 nm assuming a perfect circular cluster). Plotting the fraction of clusters with shape factor = 1 compared to all clusters revealed a significant drop in this population at later time points after stimulation. At earlier time points (30 min), there was also lower populations of these smaller, circular Myddosomes after Aβ triggering compared to LPS triggering.

## Discussion

Here, we developed ldSPIM to study TLR4 signalling and Myddosome formation triggered by Aβ aggregates and compared to those triggered by the canonical agonist LPS. The sonicated Aβ fibrils delivered from the nanopipette-triggered Myddosome formation, demonstrating that sonicated Aβ fibrils could induce an inflammatory response. Previous work has shown that this occurs through TLR4 activation and that the kinetics of this response to Aβ is much slower than in response to LPS (*Hughes et al., 2020*). The similarities and differences in Myddosome signalling kinetics between sonicated Aβ fibrils and LPS stimulation were, therefore, then characterised. We measured the first time point when

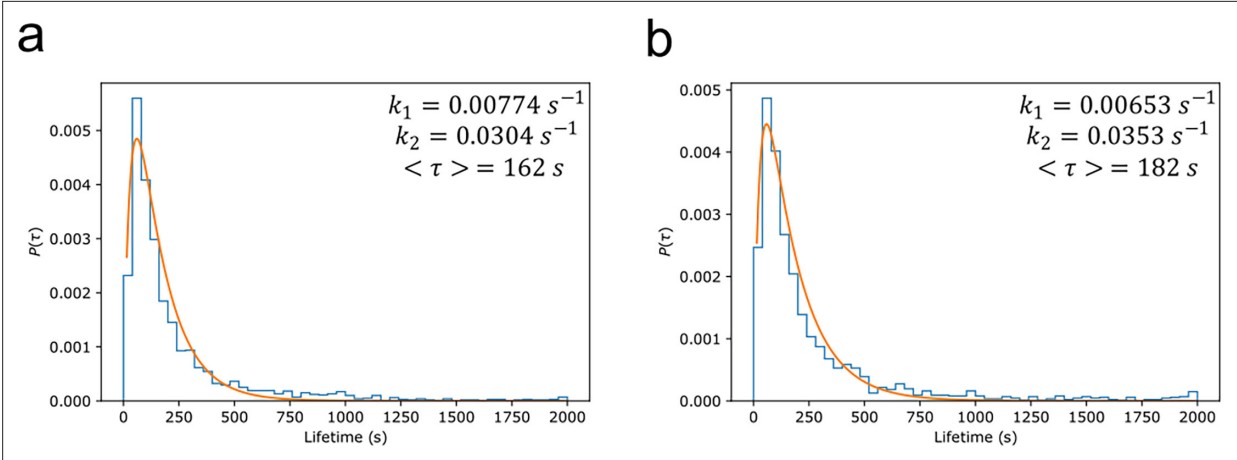

**Figure 6.** Myddosome degradation kinetics from lifetime distribution. Normalised probability distributions plotted for (**a**) lipopolysaccharide (LPS) and (**b**) amyloid-beta (Aβ) stimulated Myddosomes. The histograms were fit assuming a two-step degradation, with the rate constants $k_1$ and $k_2$, and mean lifetime $< \tau >$ calculated for each condition.

The online version of this article includes the following figure supplement(s) for figure 6:

**Figure supplement 1.** Comparison of fitting different degradation kinetics to lifetime distributions after stimulation.

a Myddosome formed, the Myddsome lifetime, the Myddosome size, and the change in the Myddosome size with stimulation time. We observed that LPS-triggered Myddosomes formed faster and are shorter lived than Myddosomes formed by sonicated Aβ fibrils. In both cases, most Myddosomes formed are removed by disassembly at the cell surface. These differences potentially support our hypothesis that the sonicated Ab fibril is multivalent binding multiple TLR4s and hence forms larger Myddosomes and these Myddosome would then take longer to form and remove.

Super-resolution microscopy of the Myddosome formed when stimulated with LPS in comparison to Aβ sonicated fibrils demonstrated a range in Myddosome size and shape. One limitation of our model is that viral transduction of MyD88 into cells may result in overexpression of the protein which could lead to structural variation that may be seen under physiological conditions. The heterogeneity in Myddosome structure and function, however, may also be functionally important. In particular, small round Myddosomes, less than 100 nm in diameter, were formed in high proportion at early times and then decreased in proportion with time. Despite the wide range of Myddosome sizes we observed, those formed by LPS triggering were on average smaller than those formed by sonicated Aβ. We also observed evidence that Myddosomes have different structures, especially at early times. We do not know, however, which of all these Myddosomes are signalling competent. LPS-triggered Myddosome signalling has a negative feedback control system whereby a short MyD88 isoform MyD88s is upregulated. Myd88s lacks the intermediate domain (ID) for IRAK recruitment and thus is signalling incompetent so this may result in the formation of Myddosomes with different structures at later times. In our previous experiments measuring Nuclear factor-kappa B (NF-κB) translocation to nucleus in response to LPS we showed that translocation occurred in the first 30 min after triggering by LPS (*Latty et al., 2018*), when we observed a high proportion of small Myddosomes. It therefore seems plausible that the small round Myddosomes, formed in higher proportion at early times, are signalling competent. The larger Myddosomes formed in response to sonicated Aβ fibrils also seem to be signalling competent, since we previously observed prolonged TNFα production over 24 hr, following addition of Aβ aggregates (*Hughes et al., 2020*), during which time we observed mainly larger Myddosomes. Determining the signalling competency of different sized Myddosomes complexes is an important goal for future studies, but the structural diversity we have observed does provide a potential explanation for how diverse signalling outcomes are possible when TLR4 is activated by different agonists.

There are some limitations to this study. First, we could not distinguish between signalling competent and signalling competent Myddosomes. In future work, we could use IRAK4 or two antibodies to identify functional Myddosomes on fixed cells, as well as working with cells where the Myddosome expression levels are at physiological levels which may reduce the formation of larger Myddosomes. Second, we have only measured the delay time for Myddosome formation when triggered by LPS or

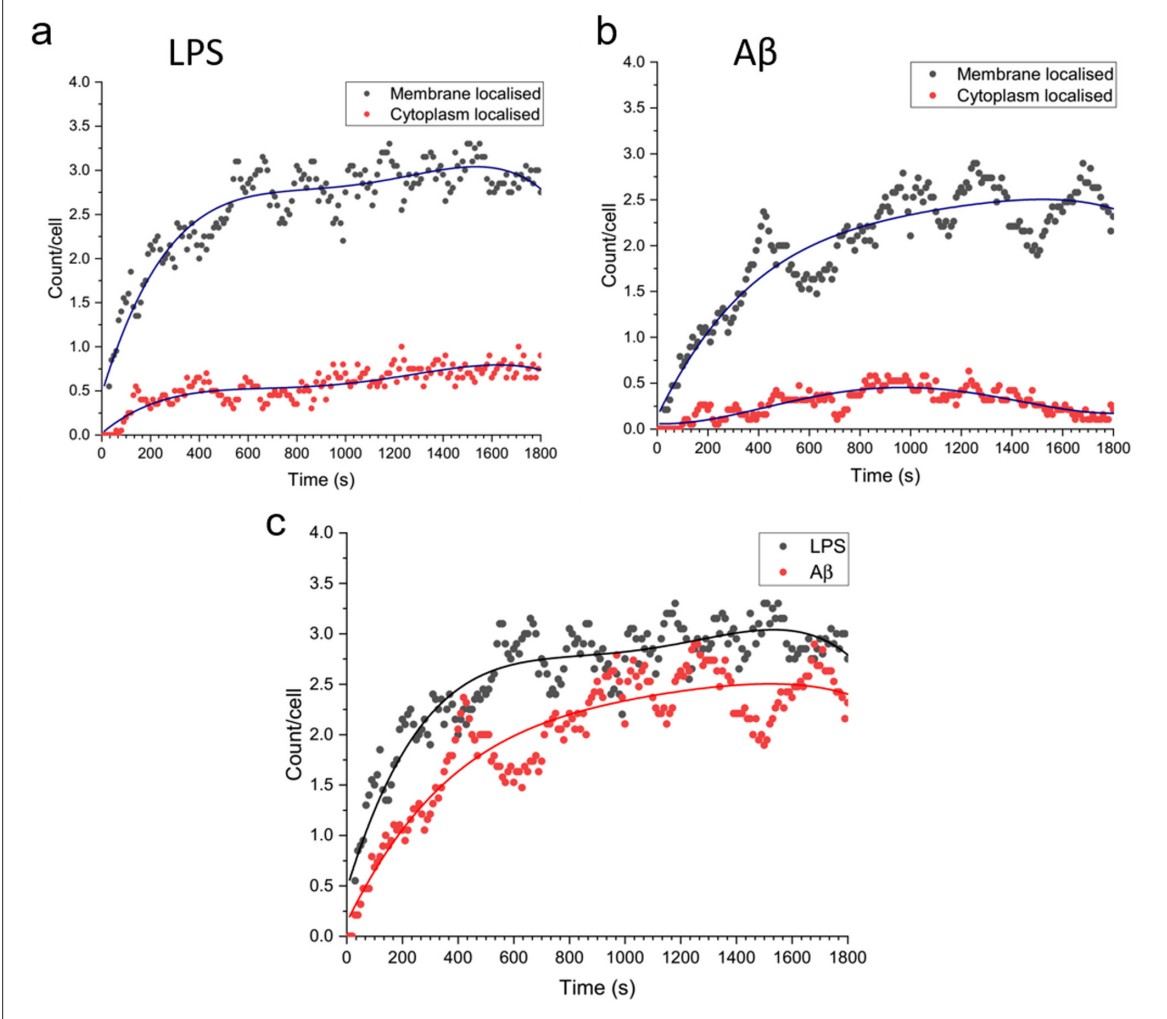

**Figure 7.** Total number of MyD88 puncta localised to the plasma membrane or cytoplasm, for Myddosomes which formed at the plasma membrane. Plots are normalised by cell number. (**a**) Lipopolysaccharide (LPS) stimulation, (**b**) amyloid-beta (Aβ) stimulation. (**c**) Overlay of total number of plasma membrane localised puncta, from membrane initialised trajectories after LPS and Aβ stimulation.

The online version of this article includes the following source data for figure 7:

**Source data 1.** Myddosome localisation (membrane/cytoplasmic) following stimulation.

Aβ aggregates. This delay times involves dimerisation of TLR4, binding of LPS or Aβ aggregates to the TLR4 dimer followed by Myddosome formation. These other processes might contribute to the difference in delay time that we observed between LPS or Aβ aggregates. It is worth noting that in our experiments we deliver the LPS or Aβ aggregates directly onto the surface for 5 s and that we previously showed the presence of the preformed TLR4 dimers on the cell surface (*Latty et al., 2018*). The affinity of Aβ aggregates for TLR4 is not known but LPS has a high affinity for TLR4, estimated to ~3 nM for lipid A–TLR4-MD-2 (*Akashi et al., 2003*). However, even with this high affinity which implies fast binding, direct delivery directly onto the surface and the presence of preformed TLR4 dimers on the cell surface we observed that it took 80 s to observe Myddosome formation. This indicates that Myddosome formation is the slow step for LPS triggering. This is likely to be the case Aβ aggregates, since pM concentrations of aggregates can trigger TLR4 signalling (*Hughes et al., 2020*) indicating high affinity. However, it is not possible to rule out a contribution of a difference in affinity to observed difference in delay time without measuring the affinity directly.

Overall, we have found that there are clear differences between the Myddosomes formed by LPS and Aβ aggregates in their kinetics, size, and shape. Furthermore, the Myddosomes formed over time also change in size and shape with continual exposure to agonist. The lifetime and structure of

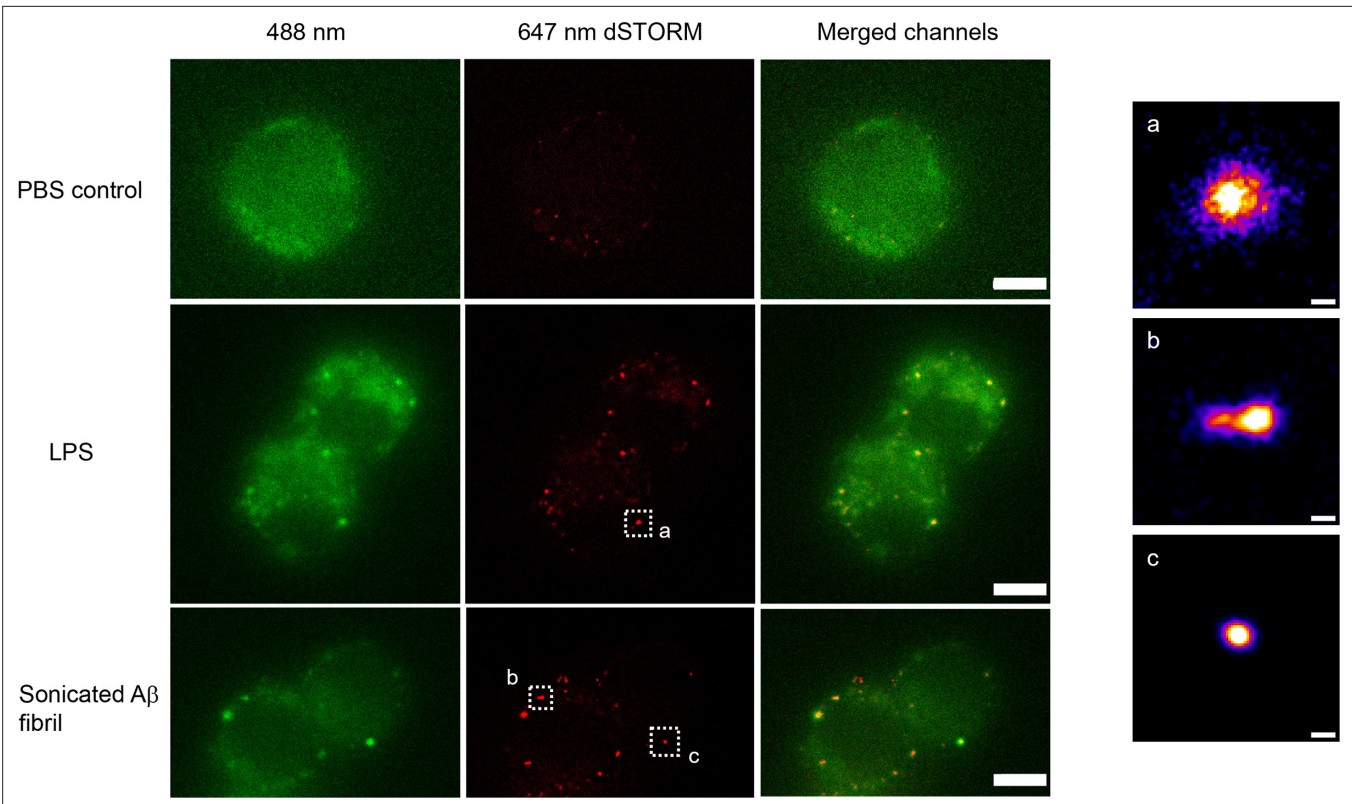

**Figure 8.** Example super-resolution images. Fixed macrophages stimulated by PBS (control), 100 ng/ml lipopolysaccharide (LPS) and 200 nM sonicated amyloid-beta (Aβ) fibrils. 488 nm channel shows the MyD88-YFP signal. 641 nm channel shows the reconstructed super-resolved image of Alexa-647-conjugated anti-GFP antibody to represent the size of Myddosome. The merged channel shows signal overlap between the Myddosomes and Alexa-647-conjugated anti-GFP antibody. Only co-localised puncta (in yellow) correspond to real Myddosome signal. The scale bars are 5 μm (left figure) and 200 nm (right).

the Myddosome formed at the cell surface appear to play an important role in the strength of downstream signalling, and the fact that the Myddosomes vary in structure and kinetics may, we speculate, play an important role in explaining how the same cellular components can give rise to multiple different cellular responses. Our experiments provide new insights into Myddosome formation triggered by LPS compared to Aβ aggregates. The combination of smaller more efficient signalling hubs, which form faster and last for a shorter time may provide a simple explanation for the larger signalling response observed with LPS than with Aβ aggregates. If this is correct, then smaller Aβ aggregates may be expected to be more potent agonists than larger aggregates. Furthermore, it suggests the reduced rate of Myddosome formation, longer lifetime and altered Myddosome structure may contribute to the weaker response from sonicated Aβ fibrils. Given the common aggregation pathway and similar aggregate structure of other aggregation prone proteins, such as tau and a-synuclein, larger multivalent protein aggregates in general may trigger TLR4 signalling less effectively than LPS giving rise to the low levels of chronic inflammation observed in neurodegenerative diseases.

## Materials and methods

### Lentiviral transfection and transduction for MyD88-YFP macrophage production

HEK293T cells and immortalised mouse macrophages MyD88$^{-/-}$ (NR-15633) were a gift from Doug Golenbock and Kate Fitzgerald now banked with BEI Resources, USA were grown in complete Dulbecco's Modified Eagle Medium (DMEM, 10% fetal calf serum, 2 mM L-glutamine, 100 U/ml penicillin, 100 μg/ml streptomycin). The media was topped up with fresh L-glutamine every 4 weeks. The lentiviral transfection and transduction needed 9 days:

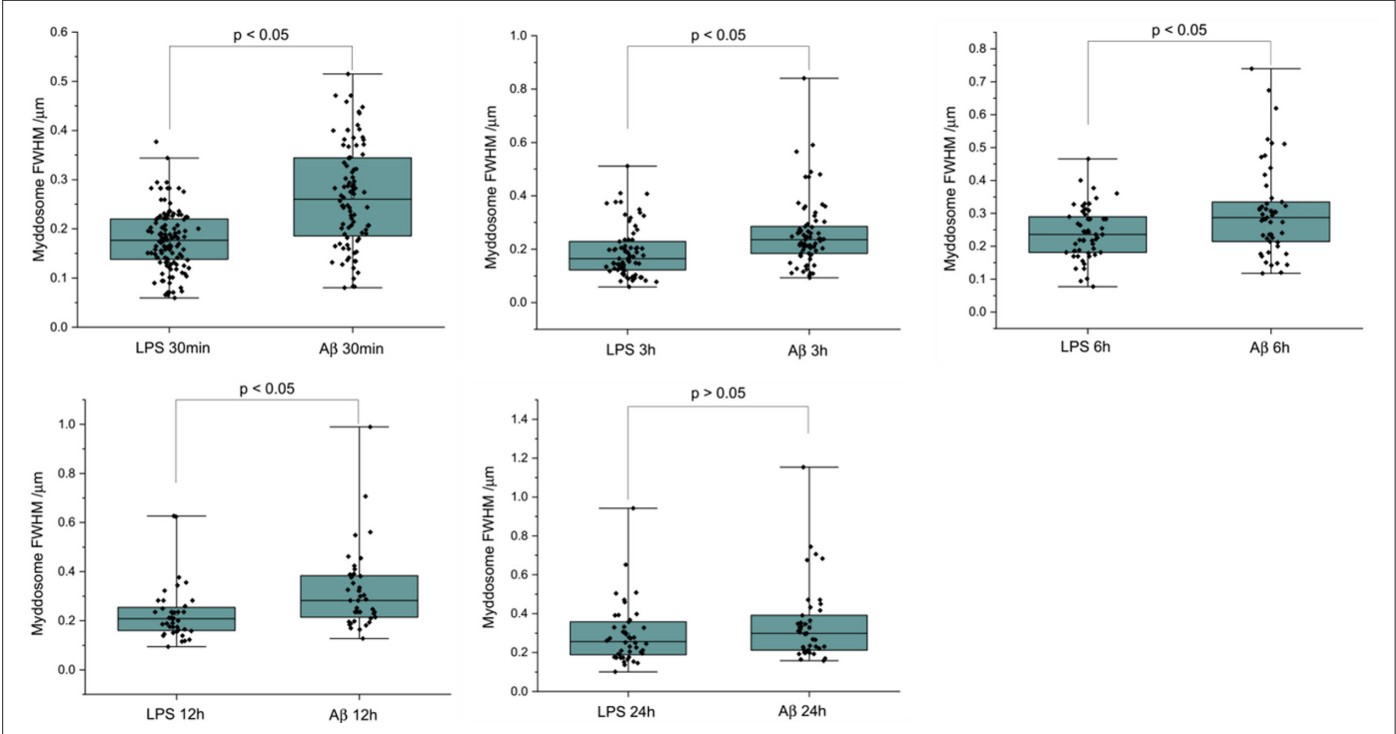

**Figure 9.** The size of Myddosomes triggered by lipopolysaccharide (LPS) and sonicated amyloid-beta (Aβ) fibrils at different times post triggering. MyD88-YFP transduced iBMDMs were stimulated with LPS (100 ng/ml) or sonicated Aβ fibrils (200 nM) followed by fixation and antibody labelling. Full width at half maximum (FWHM) of puncta was measured by 1D Gaussian fitting. (From 30 min to 24 hr: $n$ = 212, 140, 99, 77, 85 across three biological replicates for each stimulation. The p-values are based on unpaired two-sided Student's $t$-test.)

The online version of this article includes the following source data for figure 9:

**Source data 1.** Myddosome size (full width half maxima) at different timepoints following stimulation.

Day 1: The immortalised MyD88$^{-/-}$ macrophage cell line was seeded in a T75 flask.

Day 2: HEK293T cells were seeded in a T175 flask.

Day 4: HEK293T cells were ready to be harvested. The original supernatant was removed, followed by addition of 6 ml of trypsin. The cells were incubated at 37°C/5% $CO_2$ for 5 min then centrifuged at 1000 rpm for 5 min at room temperature. The cell density was diluted to $0.5 \times 10^5$ cells/ml, and 1 ml then added per well in a 12-well plate.

Day 5: 1 ml of the immortalised MyD88$^{-/-}$ macrophage cells were plated at $0.25 \times 10^5$/ml in a 12-well plate. Genejuice and DNA were added to the HEK293T cells, and then incubated for 72 hr.

Day 8: The supernatant containing lentivirus in HEK293T wells were clarified and added into wells containing the immortalised MyD88$^{-/-}$ macrophage cells and incubated overnight.

Day 9: The viral supernatant in the wells containing the immortalised MyD88$^{-/-}$ macrophages were replaced with fresh complete media. The MyD88$^{-/-}$ YFP macrophages were checked by fluorescent microscopy before use.

## Nanopipette fabrication

Nanopipettes were fabricated from quartz capillaries, with an outer and inner diameter of 1 and 0.5 mm, respectively. A laser pipette puller (Model P-2000, Sutter Instrument, CA) was used for fabrication, using the following parameters: Line1 Heat = 350, Fil = 3, Vel = 30, Del = 220, Pull = 0; Line2 Heat = 400, Fil = 2, Vel = 20, Del = 180, Pull = 255, resulting in a pipette with an internal diameter of 200 nm (±20 nm), Heat = 400, Fil = 4, Vel = 30, Del = 200, resulting in a pipette with an internal diameter of 800 nm (±200 nm). The size of the nanopipette was calculated from its electric resistance in PBS buffer; a tip size around 100 nm will generate an electronic resistance around 150 MΩ and a tip size around 800 nm will generate an electronic resistance around 20 MΩ. The 800 nm tip size was large enough to prevent blockages during Aβ delivery.

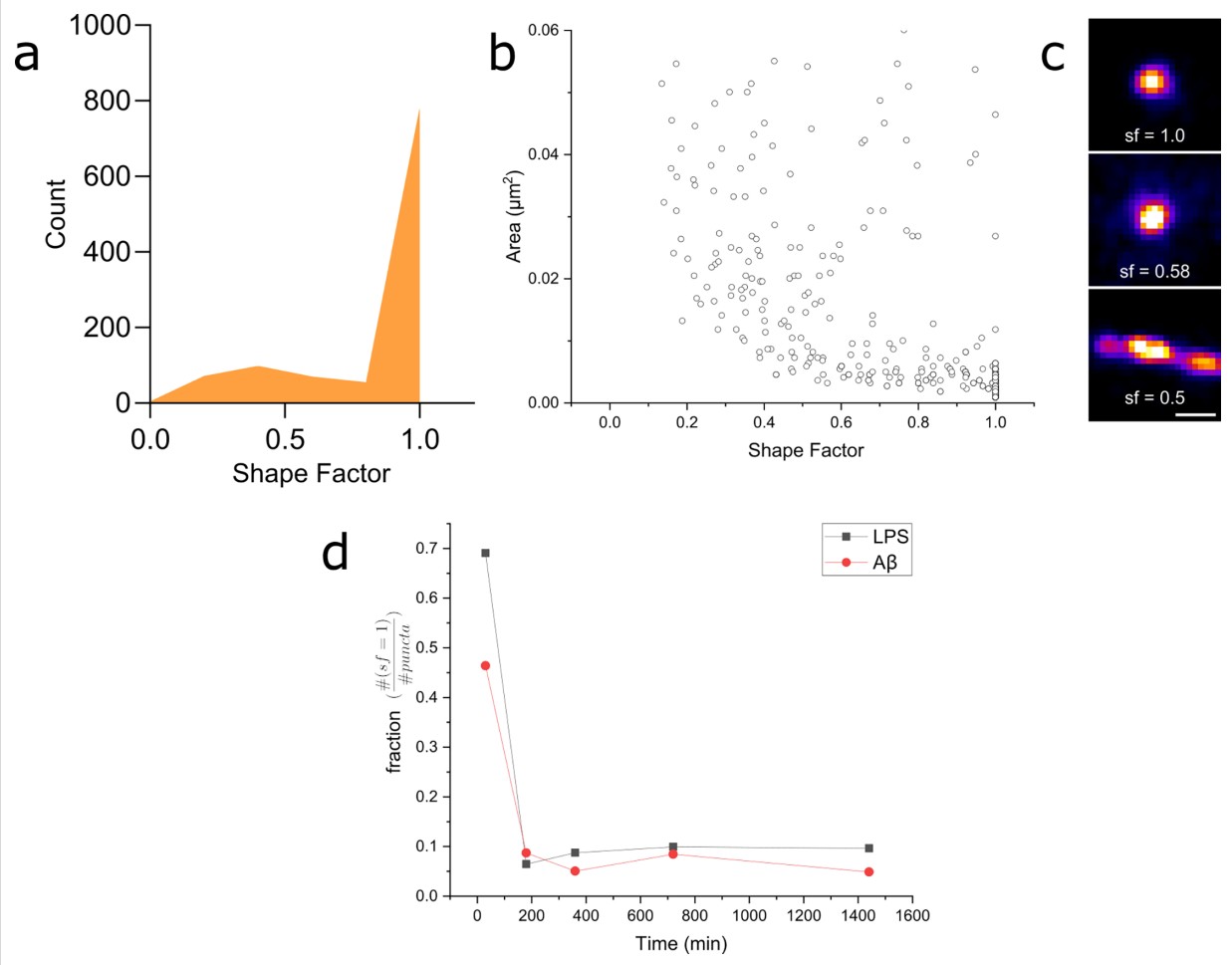

**Figure 10.** Myddosome shape factor analysis. (**a**) Distribution of super-resolved MyD88 puncta shape factors after 30 min lipopolysaccharide (LPS) stimulation (shape factor $= \frac{4\pi A}{p^2}$, where $A$ and $p$ are area and perimeter, respectively), $n = 1086$ across three independent experiments. Similar distribution is also observed for 30 min amyloid-beta (Aβ) stimulation. (**b**) Plot of super-resolved cluster area vs cluster shape factor for 30 min LPS stimulations suggests that this corresponds to a populations of very small, spherical MyD88 clusters. Similar relationship was also observed for 30 min Aβ stimulation. $n = 1086$ across three independent experiments. (**c**) Magnified examples of three super-resolved MyD88 clusters with a range of different shape factors. Scale bar is 150 nm. (**d**) Plotting fractions of spherical clusters (shape factor = 1) compared to total puncta suggest a drop in the population of small, spherical MyD88 clusters after 30 min for both LPS and Aβ stimulation. There is a smaller fraction of these small spherical clusters following Aβ stimulation compared to LPS.

The online version of this article includes the following source data for figure 10:

**Source data 1.** Raw data for Myddosome shape factor analysis.

## Amyloid-β fibril preparation

The amyloid-β fibril preparation was a modification of a method reported by our previous work (*Hughes et al., 2020*). The Aβ (1–42) peptide (Stratech, A-1163, 0.5 mg) was initially dissolved in 100% hexafluoroisopropanol (Sigma-Aldrich) at a concentration of 4 µM. This solution was incubated at room temperature for 1 hr. Then, the solution was sonicated for 10 min in a water bath sonicator and then dried under a light stream of nitrogen gas. Dimethyl sulfoxide was added to the peptide, which was incubated at room temperature for 10 min with gentle mixing. Finally, this solution was aliquoted and stored at −80°C. For a working solution, D-PBS (Invitrogen, UK) was added to the peptide stock solution and incubated for 2 hr at room temperature for peptide oligomerisation. The amyloid-β fibril was characterised using TEM (*Figure 1—figure supplement 1*).

## TEM negative staining

Aβ samples were adsorbed onto glow-discharged 400 mesh copper/carbon film grids (EM Resolutions) for about 1 min. Then, TEM grids were passed over two drops of deionised water to remove any buffer salts and stained in 2% (wt/vol) aqueous uranyl acetate for about 30 s. Uranyl acetate dye was drained off the TEM grid using filter paper and grids were allowed to air dry. Samples were viewed using a Tecnai G20 transmission electron microscope (FEI/Thermo Fisher Scientific) run at an accelerating voltage of 200 keV using a 20 μm objective aperture to improve contrast. Images were acquired using an Orca HR CCD camera (AMT, USA).

## Nanopipette delivery of LPS and sonicated Aβ fibrils

LPS and sonicated Aβ fibrils were delivered to the cell surface using nanopipette. 1 μg/ml LPS (Ultrapure LPS from *E. coli*, 0111:B4, Invivogen) and 4 μM total monomer of sonicated Aβ fibrils were loaded into the nanopipette. The nanopipette was controlled by a 3D manipulator (Scientifica microstar) to move down to the target cell until the nanopipette tip slightly touched the cell membrane. Then the nanopipette was withdrawn back by 3 μm away from the cell surface. At this position, a pressure pulse (3 kPa for 5 s) was used to deliver both the LPS and the sonicated Aβ fibrils to the cell surface (*Figure 1—figure supplement 2*).

## Live cell scanning and 3D reconstruction

The optical path of the microscope was fully described in our previous work (*Li et al., 2021*; *Ponjavic et al., 2018*). The scanning was achieved by moving the piezo sample stage. Controlled by field-programmable gate array, the movement of sample stage was synchronised with the camera image acquisition. Starting from the coverslip surface, the stage was moved in 200 nm increments. The total scan range was 20 μm, which was enough to cover the whole cell. The camera triggering model was set as edge trigger, in which the camera only acquired a frame when it received a 5 V TTL digital signal. When the sample stage was driven to a new Z slice, the FPGA would generate a digital TTL signal and send it to the camera in order to acquire a frame. This synchronisation enabled Z stack acquisition allowing for 3D reconstruction. The time interval between two scans was 10 s.

## Myddosome live tracking

An overview of the tracking analysis is shown in *Figure 2—figure supplement 1*. Image analysis was carried out using ImageJ (NIH) and a MATLAB script for particle tracking (*Weimann et al., 2013*). The raw images were assembled into xyzt hyperstacks, followed by background subtraction using a rolling-ball algorithm (rolling ball radius = 80 pixels) and 3D Gaussian blurring (*xyz* radius = 1 pixel). The hyperstacks were then 3D projected into the *xy* plane.

Puncta in the 3D projections were identified using ImageJ's Find Maxima plugin (prominence over background intensity = 20 counts) and isolated into a new stack for tracking. Particle trajectories were then extracted from the identified localisations, with the following parameters used to minimise the false detection of noise and minimise the number of lost tracks: (1) maximum search range (distance a localisation in a trajectory can move between subsequent frames) = 8 px (0.85 μm); (2) memory (the maximum number of frames where a spot can vanish, reappear and be considered in the same track) = 6 frames (60 s); (3) minimum track length = 2 frames (20 s). From the calculated trajectories, the lifetime of each track was extracted. Trajectories that formed in the first frame were treated as preformed MyD88 puncta; these were excluded from further analysis.

## Myddosome spatial localisation

Analysis was carried out using custom scripts written in python (https://github.com/p-sur/Myddosome-localisation; copy archived at *Suresh, 2024*). Cell boundaries and centroid coordinates were extracted from the acquired xyzt hyperstacks. Puncta were localised and tracked using a Python implementation based on the work by *Allan et al., 2023*; *Crocker and Grier, 1996*. The cell centroid coordinates at each time point were used to correct for any cell movement during the duration of image acquisition. Following trajectory assembly, the nearest distance of each localisation at each time point to the cell boundary was calculated, and used to segment the puncta as membrane- or cytoplasm-formed; a threshold of 1.4 μm was used to separate these populations. Cytoplasm formed trajectories were excluded from further analysis.

## dSTORM imaging of Myddosome

Macrophages were plated in complete media and allowed to sit down on a glass-bottom confocal dish (VWR International) overnight. The media was replaced with fresh media supplemented with either LPS (1 µg/ml) or sonicated Aβ fibrils (monomer concentration 100 nM) and stimulated for five different time periods: 30 min, 3 hr, 6 hr, 12 hr, and 24 hr. Following triggering, the cells were washed three times with PBS and then fixed (0.8% paraformaldehyde, 0.1% glutaraldehyde in PBS) for 15 min at 4°C. Following washing, the cells were stained with 6.7 µg/ml of Alexa Fluor 647-conjugated anti-GFP rabbit polyclonal antibody (Invitrogen) for 20 min at 4°C. The cells were then washed a further three times with PBS prior to imaging. dSTORM imaging was carried out with 50 mM Tris–HCl (pH 8), 0.5 mM glucose, 1.3 µM glucose oxidase, 2.2 µM catalase, and 50 mM mercaptoethylamine. The dSTORM images were acquired by a typical TIRF microscope, with the beam angle set for HILO illumination. Image stacks (2000 frames) were acquired after illumination with a 190-mW 638 laser (Cobolt 06-MLD-638, HÜBNER GmbH & Co KG) and 10 ms camera exposure for each field of view. Image reconstruction was performed using ThunderSTORM.

## Analysis of super-resolved MyD88 puncta

Following dSTORM reconstruction, true MyD88 puncta were identified by looking for co-localisation between the 488 nm (YFP) and 635 nm (antibody) channels. Following identification, the full width at half maximum (FWHM) of the puncta was measured by fitting a 1D Gaussian to the pixel intensities of the super-resolved puncta and recording the standard deviation of the Gaussian profile. The diameter of the super-resolved puncta was calculated using the relationship below:

$$\text{FWHM} = 2\sqrt{2ln2}\, d \approx 2.355d$$

The intensity of MyD88-YFP images was thresholded to generate a binary mask of Myddosomes. Region of interests were drawn around all filtered Myddosomes. The multiple ROIs generated were saved to a region file. These ROIs were then reloaded to the thresholded dSTORM Myddosomes images. This step ensured that analysis was performed only on the regions where MyD88-YFP was detected. An Integrated Morphometry Analysis (IMA) was then performed through IMA plugin running inside MetaMorph software (Molecular Devices) (*Kedia et al., 2021*). Parameters like the shape factor were computed for both the LPS and sonicated amyloid-β aggregates triggered Myddosomes.

## Acknowledgements

The electron microscopy work was performed using the facilities at CAIC (Cambridge Advanced Imaging Centre). This work was funded by the Alzheimer's research UK, EPSRC (EP/W015005/1), BBSRC (BB/V000276/1), the Royal Society, and the UK Dementia Research Institute which receives its funding from UK DRI Ltd, funded by the UK Medical Research Council, Alzheimer's Society, and Alzheimer's Research UK.

## Additional information

### Funding

| Funder | Grant reference number | Author |
| --- | --- | --- |
| EPSRC | EP/W015005/1 | Bing Li<br>David Klenerman |
| BBSRC | BB/V000276/1 | Clare E Bryant |

The funders had no role in study design, data collection, and interpretation, or the decision to submit the work for publication.

### Author contributions

Bing Li, Prasanna Suresh, Data curation, Software, Formal analysis, Investigation, Visualization, Methodology, Writing – original draft, Writing – review and editing; Jack Brelstaff, Formal analysis,

Investigation; Shekhar Kedia, Data curation, Formal analysis; Clare E Bryant, Conceptualization, Resources, Supervision, Funding acquisition, Methodology, Writing – review and editing; David Klenerman, Conceptualization, Supervision, Funding acquisition, Writing – original draft, Writing – review and editing

**Author ORCIDs**
Prasanna Suresh ⓘ http://orcid.org/0009-0001-5298-5994
Shekhar Kedia ⓘ http://orcid.org/0000-0002-9322-979X
Clare E Bryant ⓘ https://orcid.org/0000-0002-2924-0038
David Klenerman ⓘ http://orcid.org/0000-0001-7116-6954

Reviewer #1 (Public Review): https://doi.org/10.7554/eLife.92350.3.sa1
Author response https://doi.org/10.7554/eLife.92350.3.sa2

---

## Additional files

### Supplementary files
• MDAR checklist

### Data availability
All source data is included in the manuscript and supporting files.

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
