## [Editor Report · eLife assessment]

This **important** study uses a novel light sheet imaging technique to investigate how different TLR4 agonists regulate Myddosome formation. The data showing that LPS and A-beta can control the kinetics and size of Myddosome assembly are **compelling**. This paper should be of substantial interest to the innate immunity field.

---

## [Referee Report · Reviewer #1 (Public Review)]

Recognition of bacterial lipopolysaccharide by Toll-like Receptor 4 is an essential molecular event triggering inflammation and overcoming Recognition of bacterial lipopolysaccharide by Toll-like Receptor 4 is an essential molecular event in triggering inflammation and overcoming infection by gram-negative bacteria. However, TLR4 has recently been found to respond to other endogenously derived ligands. This has implicated TLR4 signaling in the development of disease pathology, for example, Alzheimer's disease, through the recognition of amyloid-beta. Intriguingly, the signaling response to these non-bacterial-derived ligands differs from that of bacterial-derived LPS, suggesting mechanistic differences between endogenous and bacterial-derived agonists. In this work, the authors set out to characterize these mechanistic differences. TLR4 signals through two large macromolecular complexes that assemble at activated receptors: the Myddosome and Triffosome. One hypothesis the authors aimed to test was that different ligands alter these signaling complexes' kinetics and nano-scale features. The authors focused on testing this hypothesis by examining the formation of the Myddosome in live cells. A significant strength of the paper is that the authors developed technological innovations to address this problem. Using a nanopipette delivery mechanism combined with light sheet microscopy, the authors could observe Myddosome signaling in the whole cell volume of live macrophages. This allowed them to accurately quantify the Myddosome number, size, and kinetics of complex formation and compare cells stimulated with amyloid-beta and LPS. The authors discovered differences in Myddosomes formed under LPS versus amyloid-beta stimulation. In general, amyloid-beta TLR4 stimulation resulted in slower Myddosome formation with altered morphology. One limitation of the work, which the authors point out in the discussion, is that they could not distinguish signaling-competent Myddosomes. Future work will be needed to understand whether these amyloid beta induced Myddosomes assembly have a similar or altered complement of downstream signaling proteins (such as the IRAK4/1 and TRAF6). Secondly, the structural basis for how TLR4 would distinguish between different radically agonists remains speculative, and will need further investigation. Nonetheless, this paper is important for the technological innovation to look at the molecular dynamics of signal transduction, a technology that could be adapted to study other receptor signaling pathways.

It is already known that the subcellular location of intracellular TLRs is important for limiting the recognition of self-derived ligands and maintaining tolerance. This work hints at another possible layer of regulation: that a cell surface TLR (TLR4) generates diverse signaling outcomes to extrinsic or intrinsically derived agonists by changing the dynamic behavior of signaling proteins. If correct (and much further work is required to understand endogenous TLR ligands better), it might suggest that the innate immune system employs the same molecular hardware but with altered kinetics to distinguish between exogenous and endogenous inflammatory signals. Thus, pathological aggregates or markers of sterile inflammation might be recognized and responded to by a specific signaling program that is defined kinetically. It will be an interesting direction for future studies to investigate whether and how diverse pathogen and endogenous inflammatory signals modulate the dynamics of signaling complexes.

---

## [Author Response]

The following is the authors’ response to the original reviews.

Public Review:Summary:In this manuscript, the authors set out to understand how different TLR4 agonists trigger Myddosome assembly and seek to examine how the potent LPS agonist induces a heightened TLR4 response. A strength of the study is that the authors employ a novel light sheet imaging modality coupled to nanopipette delivery of TLR4 ligands. The authors use this technological innovation to resolve the dynamics of Myddosome formation within the whole cell volume of macrophage cell lines expressing MyD88-YFP. The main finding is that the kinetics of Myddosome formation is slower for the weaker agonist Abeta than LPS. However, Abeta amyloids resulted in the formation of larger MyD88-YFP puncta that persisted for longer. The authors suggest the slower kinetics of formation and larger puncta size reflect how Abeta amyloids are a less efficient TLR4 agonist. Many Toll-like receptors are now known to recognize endogenous produced danger signals and microbially derived molecules. This work is the first to compare the signaling kinetics of endogenous versus microbially derived TLR agonists.Strengths:A key strength of this work is the technological achievement of imaging Myddosomes within the entire cell volume and using a nanopipette to administer ligands directly to single cells. The authors also combine this light sheet microscopy with STORM imaging to gain a super-resolved view of the assembly of Myddosomes. These findings suggest that Myddosomes formed in response to Abeta have a more irregular morphology. We conclude that these technological achievements are significant in improving our understanding of the dynamics of TLR4 signaling in response to diverse agonists. Given the limited literature on the molecular dynamics of innate immune signal transduction, this study is an important addition to the field.Weaknesses:One limitation of the paper is that a suitable explanation for how larger Myddosomes would contribute to an attenuated downstream signaling response. Do the larger clusters of nucleated MyD88 polymers reflect inefficiency in assembling fully formed Myddosomes that contain IRAK4/2? Could the MyD88-GFP puncta be stained with antibodies against IRAK4 (or IRAK2) to determine the frequency and probably of the two ligands to stimulate signal transduction beyond MyD88 assembly?A second weakness is the discussion. The authors should explore other explanations for the observed differences in Myddosome formation between TLR4 agonists. For example, could the observed delay in Myddosome assembly in response to Abeta be due to different binding affinity or kinetics to TLR4? Can this be ruled out?

We thank the reviewer for these comments.

To address the first comment we have added a section on the limitations of the current study and suggested that future work could use IRAK4 or 2 staining to identify Myddosomes that are functional as well as working with cells where the Myddosome expression levels is at physiological levels, which may reduce the formation of larger Myddosomes.

The reviewer is correct that the difference in delay time for Myddosome formation could be due slow formation of a TLR4 dimer or binding to the TLR4 dimer, rather than the time take to assemble the Myddosome after TLR4 dimerisation and binding since we have only measured the delay time for Myddosome formation when triggered by LPS or Aβ aggregates. This delay times involves dimerization of TLR4, binding of LPS or Aβ aggregates to the TLR4 dimer followed by Myddosome formation. These other processes might contribute to the difference in delay time that we observed between LPS or Aβ aggregates. It is worth noting that in our experiments we deliver the LPS or Aβ aggregates directly onto the surface for 5 seconds and that we previously showed the presence of the preformed TLR4 dimers on the cell surface (Latty et al., 2018). The affinity of Aβ aggregates for TLR4 is not known but LPS has a high affinity for TLR4, estimated to ∼3 nM for lipid A–TLR4-MD-2 (Akashi et al., 2003). However, even with this high affinity which implies fast binding, direct delivery directly onto the surface and the presence of preformed TLR4 dimers on the cell surface we observed that it took 80 s to observe Myddosome formation. This indicates that Myddosome formation is the slow step for LPS triggering. This is likely to be the case Aβ aggregates, since pM concentrations of aggregates can trigger TLR4 signalling (Hughes et al., 2020) indicating high affinity. However, it is not possible to rule out a contribution of a difference in affinity to observed difference in delay time without measuring the affinity directly.

We have added both these points to a new paragraph on the limitations of the study in the Discussion.